# Food Heritagization and Sustainable Rural Tourism Destination: The Case of China's Yuanjia Village

**Jing Guan [1], Jun Gao [1],\* and Chaozhi Zhang [1,2]** 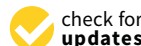

[1]  School of Tourism Management, Sun Yat-sen University, Zhuhai 519000, China;
    guanj23@mail2.sysu.edu.cn (J.G.); zhchzhi@mail.sysu.edu.cn (C.Z.)
[2]  Institute of Belt and Road Studies, Sun Yat-sen University, Zhuhai 519000, China
\*   Correspondence: gaoj63@mail.sysu.edu.cn; Tel.: +86-0756-3668279

**Abstract:** A "cultural turn" from those early management focused studies to more wholesome and exploratory socio-cultural analysis of food with sociological approaches has emerged in more recent food tourism studies. In the Chinese tourism context, however, extant studies are mainly conducted with marketing approaches linked to management to analyze the role of (heritage) foods in tourist perception and experience, and destination marketing. This study thus seeks to understand the mechanism of food heritagization and its effect in Yuanjia Village, a renowned rural tourism destination featuring traditional delicacies in China's Shaanxi Province. Data were collected via interviews and participant observations. The findings show that, led by local elites and monitored by a bottom-up regulatory system, locals use raw materials associated with being "local", "traditional", and "organic", make food with traditional crafts, and present food in a nostalgic atmosphere for consumption. Traditional foods are reinvented/reproduced as edible exemplars of the culture and heritage of the Guanzhong area (central Shaanxi Plain), and as carriers of nostalgia for a rural past that satisfies the imaginations and needs of surrounding urbanite visitors. This, in turn, contributes to the sustainability of the village as a rural tourism destination, featuring industry convergence that fosters economic sustainability, as well as governance embedded in rurality to deal with tourism benefit distribution (i.e., cooperatives) and social problems (i.e., peasant school) that promotes social sustainability. This research contributes to the understanding of food heritagization from a bottom-up perspective as well as rural destination sustainability from a gastronomical perspective in China.

**Keywords:** food heritagization; rural tourism; sustainability; nostalgia; food safety; cooperatives

---

## 1. Introduction

Food is a salient part of the tourism system. Food provision is not only about meeting tourists' physical needs, but also relates to the tourist experience as well as destination marketing [1,2]. In fact, local foods are increasingly seen and deployed as an important resource for tourism development, especially in rural areas [3,4]. Against such a backdrop, the study of the relationship between food and tourism has seen an unprecedented growth and popularization over the last decade [5]. Notably, a "cultural turn" from those early management-focused studies to more wholesome and exploratory discussions of food and culture has emerged in more recent studies [5,6]. Such a "turn" draws growing academic attention to the cultural meaning, experience, and permanence relating to food, as Timothy and Ron argued, "cuisine is, without doubt, one of the most salient and defining markers of cultural heritage and tourism" [7]. In this regard, a central theme in food tourism studies is the analysis of the impact that tourism practices have on reinvention of local foods as symbols of cultural heritage [7,8], that is, food heritagization [3].

Yet, in the Chinese context, extant tourism studies are mainly conducted with marketing approaches linked to management to analyze the role of food heritages in the tourist perception and experience, and destination marketing [1,9,10]. In fact, the wider food tourism studies with sociological approaches are skewed towards the West, particularly Europe [3,11]. Socio-cultural analysis of the link between food and tourism in China is very limited, and mostly relates to a macro-scale political economy context, emphasizing the role of the Chinese state and/or market forces (e.g., capitalistic enterprise, tourist demand) in the process of food heritagization [3,12,13]. Regarding food heritagization, two contextual factors in China stand out from those of the West [12]: First, China's official and public discourses have paid little attention to the conservation roles fulfilled by the marketing of local/peasants' foods, such as their importance in saving traditional heirloom seeds and traditional farming methods; second, the marketing of local/peasants' foods has been promoted largely by the government (through, for example, tourism bureaus and official "New Countryside" programs and campaigns), rather than by grassroots initiatives. Furthermore, the literature on heritagization of food tends to dichotomize the interests of agents, between those of government, corporate, or elite entities on one hand, and local or regional communities on the other [14]. To this end, the food heritagization process on a local level in the Chinese tourism context and its subsequent impact remain almost unexplored [15], albeit that food heritagization is frequently envisaged and promoted as a tourism development strategy.

To this end, this study posits food heritagization as a social construction [16], emphasizing the underlying processes by which various agents articulate certain foodstuffs as heritage in an attempt at pursuing their respective aims [17]. It should be pointed out that such a critical approach towards food heritages [18] has been adopted in many previous studies, such as Klein's study on local cheese heritagization in China's Yunnan [3], Grasseni's study on local cheese in the northern Italian Apls [19,20], Littaye's study on pinole (a Mexican traditional sweet) [17,21], etc. The current study seeks to understand the mechanism of food heritagization and its development consequence in China's Yuanjia Village, a renowned rural tourism destination featuring traditional delicacies. More specifically, this study probes into actors' discourses and practices of re-producing or re-inventing traditional foods as heritage on a local level, and how such re-production or re-invention affects the development of the village as a rural tourism destination. It contributes to a nuanced understanding of food heritagization on a local level as well as rural tourism destination sustainability from a gastronomical perspective in China.

## 2. Literature Review

### 2.1. Food and Rural Tourism

Food is nowadays closely related to rural tourism, which is increasingly conceptualized as cultural consumption of a rustic and idealized rurality, along with a view that food with traditional and local characteristics can be reinvented as a tourism product or tourist attraction [22]. In practice, utilizing food for rural tourism development has been the concern of policymakers, advisors, entrepreneurs, and researchers worldwide [22–25]. In the Chinese context, rural tourism started with nongjiale (farmhouse joy) in the 1990s in the suburbs of cities, for which eating local peasants' foods is the most important [12]. The nonjiale movement has gained momentum since the 2000s and has penetrated all through China's countryside, including ethnic minority areas far away from cities, which has been largely driven by new consumption orientations and needs [12]. On the one hand, with China's rapid development during the Reform era, a "retro-boom" has emerged among middle-class urban residents, seeing a collective nostalgia for rural life to confirm their current modern identity amid and against the environmental risks generated by the on-going urbanization and modernization process [12,26,27]. Peasants' foods or local foods of the rural are thus often reproduced as symbols of rurality and rusticity to meet such a urban demand [12], which have become a core selling point of rural tourism in China [28]. Indeed, as Park [27] pointed out, "China's countryside in general becomes

an appealing place of tourism, a destination saturated with new bundles of cultural meaning and value that constitute China's rurality today".

On the other hand, due to the industrial-oriented food production methodology, featuring the pursuit of mass production and the addition of various chemical substances, foods gradually lose their original taste, and may even contain harmful substances [29]. Notably, frequent food safety incidents have filled people with mistrust and panic about industrially produced foods [30]. This is especially the case in China, where a series of poisonous food scandals (e.g., poisonous infant formula) since the early 2000s have "haunted" ordinary citizens, causing public mistrust in its modern food system [31]. In this regard, China's rural–urban divide has profound implications for peasants' foods or rural foods. Specifically, in dominant public discourse, the rural, as the opposite of the urban, is often imagined as being "traditional", "moral", "backward", "ecological", etc. [27,32]. Hence, urban agro-tourists tend to romanticize the peasantry's traditions and "simple" foods, yet simultaneously castigate farmers as backward, ignorant, and unhygienic [27]. It is in such a context that peasants' or rural foods reproduced as being "traditional", "healthy", "local", "authentic", etc. are gaining popularity [12]. It should be noted, however, that the new consumption orientations and needs towards food resulted from modernization is not a phenomenon limited to the Chinese context. As a matter of fact, in many places around the world with emerging middle class populations who remember (or reimagine) the food of their former rural or less urbanized lives, a "gastronostalgia" has fostered the popularity of traditional foods [33,34].

## 2.2. Food Heritagization and Sustainable Rural Tourism Development

Thanks to the growing importance of food in rural tourism, the packaging and reproduction of traditional cuisine, or food heritagization, has been envisaged and promoted as an effective action for local development [31]. Indeed, it is often argued that local food perceived as authentic and linked to local culture and heritage would work as an effective tool to sustain rural tourism particularly, and rural communities generally [23,24]. Yet, food heritagization is far from a technical development process, but is a contested and negotiated social process in which various actors seek to articulate certain foodstuffs as heritage for their own benefits [16,17]. Therefore, the access to opportunities that emerge from food heritagization may not be equal, and the question of who can capture value from heritage foods deserves close examination [3]. In this regard, many cases illustrate that the market value created by food heritagization has been reaped by powerful, extra-local actors, even in the West [11], thus putting forward the question of how heritage foods can contribute to the sustainability of the rural tourism destination in question.

In China, food heritagization has much relevance to state projects of agricultural modernization that are envisaged as an important strategy to revitalize its decaying rural areas in the rapid urbanization and modernization process [3,35]. As a matter of fact, proactive state intervention, often in alliance with market capital, has been noted as a distinctive feature of food heritagization in China [3,12]. Rural communities, however, are never passive actors, albeit extant studies tend to focus on the macro-structural forces of Chinese state and market in shaping heritage foods [3]. One should point out that heritage food differs from other forms of heritage in that it often originates from peasants'/smallholder farmers' everyday life, and is of close relevance to their livelihood and identity issues [3,36]. To this end, to examine on-the-ground actors' role in food heritagization and its effect on them, that is, a bottom-up approach to understanding food heritagization is of significance to understand how China's food heritagization may better contribute to the sustainability of rural tourism particularly, and rural communities generally.

## 3. Research Design

### 3.1. Research Context

Yuanjia Village, located in Liquan County of central Shaanxi province, is only 60 kms away from the provincial capital Xi'an city (Figure 1). The village covers an area of 0.33 km$^2$, with 286 people in 62 households. Since the launch of rural tourism in 2007, the village has seen great changes, becoming a popular rural tourism destination nationwide. It was listed as a National AAAA Scenic Spot in 2014 as well as a National Rural Tourism Demonstration Village in 2015. The whole village has been marketed as the place to experience the Guanzhong area (central Shaanxi Plain). In 2017, the village received 5.6 million tourists, its tourism revenue reaching RMB 380 million (amounting to US$57 million). Also, over 5000 locals from the village itself and surrounding villages are currently working or doing business there. It should be noted that whilst the village has a national reputation, most of its tourists come from the surrounding big cities, such as Xi'an, Xianyang, Zhengzhou, etc.

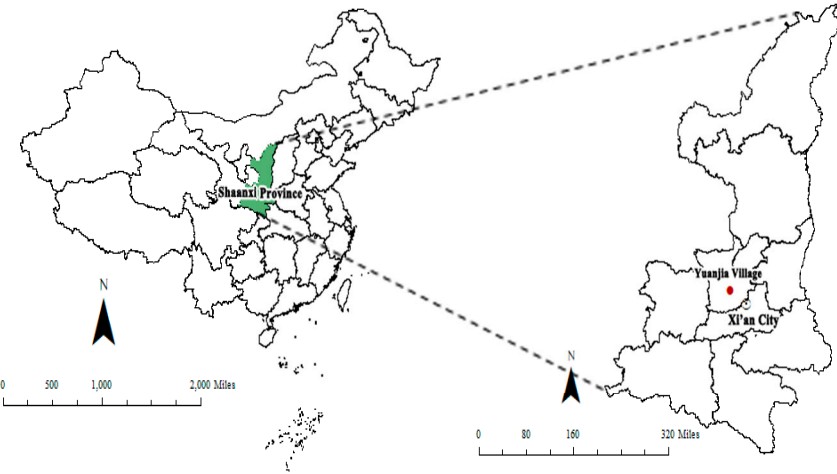

**Figure 1.** Location of Yuanjia Village in China.

Snacks are the core attractions of the village, and its snack street is a must for visitors to the village (Figure 2). In 2017, the total income of the snack street reached RMB 150 million, accounting for 40% of its annual tourism income, with daily turnover exceeding RMB 2.0 million. Apart from the food business, other tourism businesses are also emerging, such as homestays, tea houses, bars, creative studios, parent–child activities, etc., making the village a comprehensive rural tourism destination.

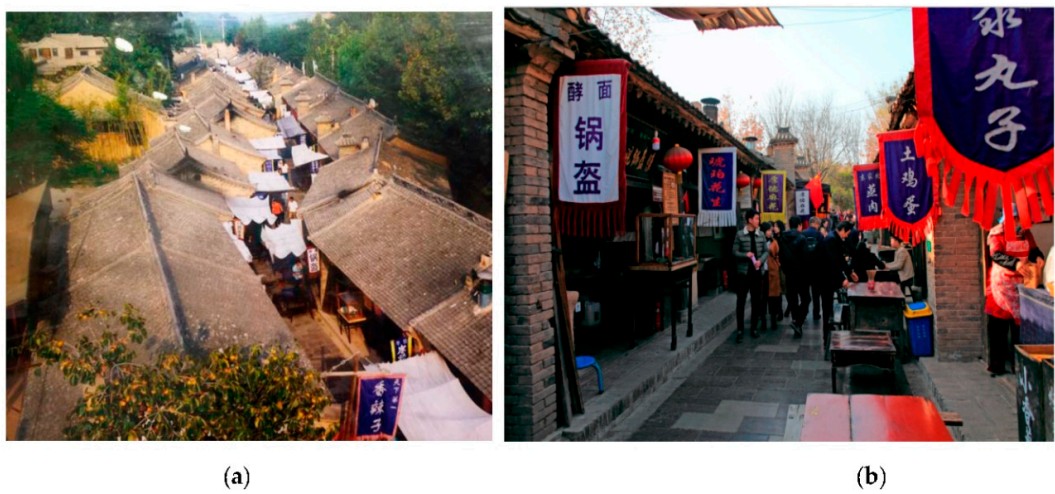

(**a**)                       (**b**)

**Figure 2.** Snack street. (**a**) Aerial view of snack street; (**b**) Snack street scene.

## 3.2. Data Collection

Given the exploratory nature of the research, a qualitative approach was adopted. For its part, participant observation and interview were adopted as the different sources of data that could triangulate, as well as complement, each other.

Participant observation: The first author made three visits to the site to specifically conduct this research, in September 2017, December 2017, and January 2018, respectively, amounting to 22 days of stay. While staying at the village, the first author repeatedly observed the street characteristics, food production, and presentation during the investigation. This also included observing tourist activities, as well as host–guest interactions in different situations, such as tourists inquiring about food ingredients and taste at snack bars. Notably, the first author occasionally worked as a volunteer tour guide to show tourists around the village, which permitted him to interact directly with tourists to interrogate their views of the village, especially its snacks. Moreover, he participated in the "Yuanjia Village Rural Tourism Training Class" (between 21 and 22 September 2017), the weekly meeting of the villagers (on the evening of 25 September 2017). Field notes were taken.

Interview: Thirty-six face-to-face interviews were conducted with village cadres, food operators, ordinary villagers [1], and tourists (Table 1), in formal or informal ways. Data saturation was reached, and large amounts of information emerged from the interviews, triangulating each other [37]. For formal interviews, they were conducted with semi-structured questions. The interview questions were mainly related to the development process of the snack street, food production process, tourist experience, management, and operation of tourism at the village level. Purposeful sampling was adopted to cover a wide range of respondents, as well as these key figures in local tourism development. These interviews lasted between 15 min to 2.5 h. All were recorded with respondents' permissions and later transcribed. For informal interviews, they were often conducted out of convenience with a conversational approach, such as causal talks over mealtime, which allowed more natural responses. Also, it should be noted that contacts with key respondents were maintained via Wechat (an instant messaging application widely used in China) after the fieldtrips, which allowed further investigation when new questions emerged during data analysis.

**Table 1.** Respondents Profile.

| No. | Gender | Age | Identity Information | No. | Gender | Age | Identity Information |
|---|---|---|---|---|---|---|---|
| L-01 | Male | 40+ | Party secretary of the village | T-01 | Male | 30+ | From Shaanxi |
| L-02 | Male | 50+ | Village elite | T-02 | Male | 40+ | From Shaanxi |
| L-03 | Male | 50+ | Chilli workshop owner; village elite | T-03 | Male | 40 | From Jiangsu |
| L-04 | Male | 50+ | Deputy director of the village committee | T-04 | Male | 30 | From Shaanxi |
| L-05 | Male | 34 | Deputy director of the village committee | T-05 | Male | 35+ | From Henan |
| L-06 | Male | 33 | Office director of the village committee | T-06 | Male | 30+ | From Shaanxi |
| L-07 | Male | 26 | Ordinary employee of the village | T-07 | Male | 40+ | From Shaanxi |
| L-08 | Female | 40+ | Vermicelli shopkeeper | T-08 | Male | 35+ | From Shaanxi |
| L-09 | Female | 50+ | Sheep blood soup shopkeeper | T-09 | Male | 30 | From Shaanxi |
| L-10 | Male | 45 | Tofu shopkeeper | T-10 | Female | 40+ | From Gansu |
| L-11 | Male | 50+ | Youtuotuo(a local snack) shopkeeper | T-11 | Female | 20+ | From Yunnan |
| L-12 | Male | 60+ | Sanzi(a local snack) shopkeeper | T-12 | Female | 35+ | From Sichuan |
| L-13 | Male | 33 | Handicraft shopkeeper | T-13 | Female | 26 | From Shaanxi |
| L-14 | Female | 30 | B&B owner | T-14 | Female | 30+ | From Shaanxi |
| L-15 | Male | 60+ | Ordinary villager | T-15 | Female | 25+ | From Henan |
| L-16 | Male | 60+ | Ordinary villager | T-16 | Female | 30+ | From Shaanxi |
| L-17 | Male | 40+ | Ordinary villager | T-17 | Female | 30 | From Shaanxi |
| L-18 | Female | 50+ | Ordinary villager | T-18 | Female | 24 | From Fujian |

## 3.3. Data Analysis

Thematic analysis was adopted to analyze the qualitative data. First, all transcripts were read word by word, and open codes were obtained while certain words or sentences were highlighted. Second, the open-coded quotations related to the research objectives were taken out and read word by word with reference to field notes to identify sub-themes. Third, the quotations relating to identified

sub-themes were interrogated to identify themes. Last but not least, communication was maintained with key respondents via WeChat to double check the analysis.

## 4. Findings

### 4.1. Food Heritagization on the Ground

#### 4.1.1. The Initiative of Using Traditional Food as Attraction

Yuanjia Village is an ordinary small village, without obvious advantages in tourism resources. When it began to develop rural tourism in 2007, it mainly relied on nongjiale (farmhouse joy) and the Folk Street (displaying folk customs). "Secretary Guo's [2] plan is to attract tourists through the Folk Street, sell what is made in Folk Street's workshops to tourists, and then provide basic food and accommodation services for tourists through nongjiale. In Guo's imagination, the countryside should be different from the city, and he wanted to present his childhood memories of the village" (L-02). This mode of development gained some success, and in 2011, the village leader decided to build a snack street to provide traditional snacks for tourists to promote further development. "At that time, seeing the popularity of nongjiale and the Folk Street, Guo considered that the business still had to focus on 'eating' for development in the future. Since many people came to the countryside to pursue 'eating', it should be good business" (L-03).

Guo had observed these business opportunities, so he decided to re-adjust the development of rural tourism by reproducing the "traditional taste" and "childhood taste" of the Guanzhong area (central Shaanxi Plain). "When we 'attract' operators, we ask these people to be able to make 'old tastes'. Yuanjia Village wants to attract city people with the traditional tastes" (L-03). "Each snack in Yuanjia Village is made in accordance in traditional ways by local peasants. They can make the taste of the countryside, the taste of home, the taste of nature and the taste of culture. If you keep the old taste, you will naturally be able to retain the tourists" (L-01). Hence, the initiative of using traditional food as the attraction of Yuanjia Village is, to some extent, attributed to the village head's development vision. A shift from peasants' foods (provided by nongjiale) to traditional foods (sold at the Snack Street) also marked the beginning of food heritagization on the ground, as these foods have been constructed as edible exemplars of Guanzhong culture and heritage since then, which will be elaborated in the following section.

#### 4.1.2. Reproducing Traditional Food: Locals' Perspective

Food materials: Regarding food materials, locals not only pay a lot of attention to the quality, but also emphasize the origin of the materials. Locally-produced and traditional materials are much valued. The deputy director of the village committee (L-04) commented: "All the ingredients are mainly produced locally, such as wild vegetables. There are different kinds of wild vegetables in different seasons, and they are collected from fields or mountains by peasants around".

When locally produced materials cannot meet the demand of the food sector, the village chooses ideal production sites elsewhere to produce food materials in order to ensure the quality of food materials. For example, the village established wheat and chili plantations in the neighboring Weinan city and Xingping city of the province. Notably, to produce materials with traditional methods is also highlighted, as the chili workshop owner (L-03) noted: "I would buy at least 200 tons of high-quality chili a year. I ask growers at the pepper base to adopt a harmless approach to pest and disease management. The method is to mix the plant ash with water and spray it on the pepper seedlings after soaking. This ensures that the pepper is free of any pesticide residues". Similarly, as one village elite (L-02) noted: "Nowadays, a lot of pastas don't taste as good as they used to be, since the wheat seeds have changed. Yuanjia Village use traditional seeds to grow wheat. Although the yield is low, the quality is good, it can ensure the traditional taste". To this end, the food materials are often associated

with being local, traditional, and organic, thus ensuring the quality of traditional foods sold to tourists from the beginning.

Food making: Foods are produced in traditional ways via using traditional material processing technics and selecting producers with traditional craftsmanship. For example, to make chili oil, locals use stone mills instead of machines to grind chili. In this regard, as the chili workshop owner (L-03) explained, "You would find that the surface of the chili processed by the machine is irregularly shaped under microscopes, and the edges are all quite sharp. However, the surface of the chili ground by a stone mill is smoother under the microscope. Although you can't see it with your eyes, when you eat it, you will find that there is a huge difference in taste between them . . . . These are all the old stone mills I collected from the countryside. Some of them date back to the Qing Dynasty (1644–1912) and Republic of China period (1912–1949)".

When it comes to recruiting food producers, the village would conduct a serious inspection of each person and set food making competitions. "The village would find several highly respected people to judge who makes the best snacks. In the end, it would be agreed that the person who makes the best food would be selected. This ensures the best craftsmanship and taste of the snacks" (L-04). In fact, some of the traditional food making crafts, as well as traditional foods, have been designated as intangible cultural heritage at the provincial or municipal level. Also, the food making processes are open to visitors to gaze on.

Food presentation: Food utensils and dining environment are also key elements in the construction of traditional food [38]. Snack shops are decorated with characteristics of a traditional rural kitchen, such as square tables, small stools, benches, and thick porcelain bowls. Moreover, food producers and workers are required to dress up in a traditional "rural" way: "Store owners and service workers in snack streets must wear rustic jackets and trousers. There are no specific uniform requirements for shopkeepers in terms of color and style; however, clothes must have the characteristics of the countryside. This often means that they cannot wear clothes that are too modern. This can give the guests a sense of authenticity, let the tourists know that this is the traditional food made by local peasants" (L-06). These measures thus help to create a traditional rural dining atmosphere (Figure 3).

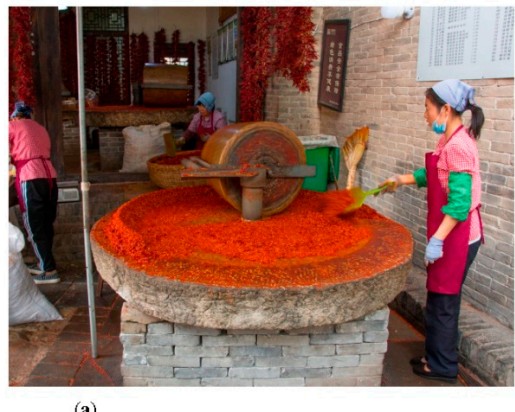 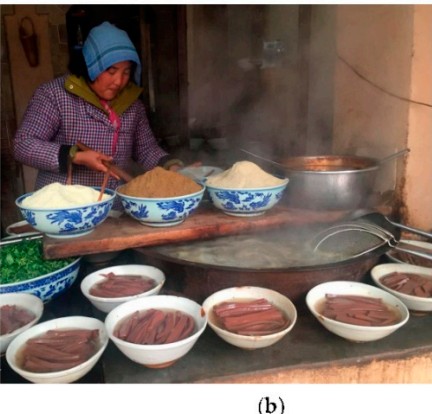

(a)　　　　　　　　　　　　　　　　　　　　　　　　　　　　(b)

**Figure 3.** Food Production Scene. (**a**) Chili powder ground by stone mill; (**b**) Sheep blood soup ③.

*4.2. Bottom-Up Regulatory System of Traditional Food Reproduction*

4.2.1. Daily Supervision of Food Quality by Village Committee

Most raw food materials are purchased by the village committee and then distributed to food operators. For example, rice, soybean, sweet potato, and chili, which are in high demand, are supplied by cooperative planting bases or companies designated by the village committee. The village does not allow food operators to purchase food materials on their own without authorization, and those who violate the regulations would be disqualified as food operators. As such, through unified procurement, the source of materials can be traced to ensure food safety.

The village committee also plays an active role in supervising food processing. L-04, the deputy director of the village committee, believes that "Food quality problems are most likely to occur during processing, so we will strictly control the food processing process. For each of our snacks, we will follow the traditional process and not use any additives". Hygiene is of great importance, and one must abide by related regulations. L-04 noted, "Chopsticks in each snack bar are required to be changed regularly. Tableware should be disinfected and if tableware is found to be used without disinfection, the snack bar will be closed for three days". In this aspect, as "food safety supervisor", each village committee member is in charge of one street, inspecting the snack bar once a day.

### 4.2.2. Supervision by Sector Association

Members of a self-elected association of the snack street inspect the food from time to time, which is also an effective force in monitoring the daily operations of snack bars. The followings are some typical comments:

"We'll put the owner out of business if he breaks the rules, for example, by using something of poor quality. When the snack street first opened, we drove away an operator because the quality of his snacks was not up to standard" (L-03).

"The association generally carries out inspections from time to time. It not only checks the source of raw materials and freshness of food, but also inspects the tableware layout, store environment and clothing. If problem is found, the shopkeeper will be asked to rectify it immediately, and if the situation is serious, then the shopkeeper will be criticized at the weekly meeting of the village" (L-05).

### 4.2.3. Regulation of Local Authority

The Food and Drug Administration of the county is the main government agency responsible for food production supervision of the whole county. It has played an important role in monitoring food production at the village, conducting various inspections and implementing related regulations. "The Food and Drug Administration conducts spot checks on raw materials at the village once a quarter, while a commissioner of the agency is deployed at the village to inspect food production on a daily basis. There are also other types of inspections on a weekly or monthly basis, which helps to ensure food safety" (L-05). Besides, the Administration requires food operators to study relevant laws and regulations, such as the Food Safety Law of China. Various food safety warning signs can be seen in the village.

### 4.2.4. Peasants' Moral Self-Discipline

A moral self-discipline approach is also adopted. Food operators hang "vows" to promise the "authenticity" and safety of their snacks (Figure 4). "We didn't require each snack bar to have an oath sign. At first, only one snack bar had an oath sign. Later, it was found that businesses with an oath sign attracted more customers, so now more snack bars hang vows" (L-04). Snack bar owners believe that the swearing on food safety would keep them in awe when making food. "The fact that I have taken such an oath shows that I have confidence in the quality of the food I make, and I believe that there will be a God watching over me, so I would not mess with the cooking process" (L-07). Such a self-discipline approach is regarded as effective in fixing the loopholes of other supervision systems to completely dispel tourists' food safety concerns, reflecting the village's slogan "Farmers to Protect Food Safety" (Figure 5).

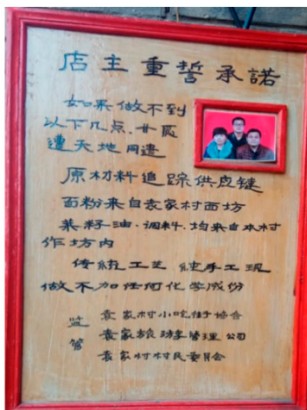

**Figure 4.** Vow of Snack Shopkeeper ④.

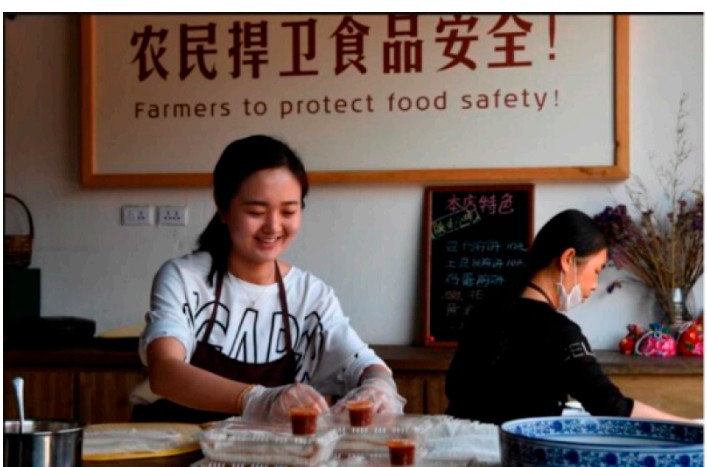

**Figure 5.** Farmers Safeguard Food Safety.

*4.3. Tourists' Consumption of Traditional Food*

For urbanite tourists, the traditional foods arouse a strong sense of nostalgia about a beautiful "past": "The taste of foods ate during childhood at hometown would always stay in our memories, which motivates people to search for such a taste ... The snacks here enables me to experience the taste in memories" (T-06). "The snacks sold in Yuanjia Village may not look so exquisite, but they are reminiscent of things made at home when you were a child, like the steamed bread without any additives. It doesn't look so white but tastes the same as the steamed bread eaten at home when one was a child" (T-07).

Also, the constructed traditional rural dining atmosphere triggered a sense of authenticity associated with food. For example, one tourist (T-10) noted, "The aunts (a common way for Chinese people to refer to women older than themselves) who sell snacks dressed in the kind of coarse cloth that rural women used to wear, which seems to be the kind of people who are kind and make very authentic food".

Last but not least, the associations of food with local peasants, such as the aforementioned moral self-discipline regarding food safety, appears to enhance tourists' trust in food safety at the village. One tourist (T-05) said, "This slogan (Farmers to Protect Food Safety) is very good. It reminds people of the simplicity and kindness of farmers. Peasants do not add food additives or other things to their food". In a similar vein, another tourist (T-03) commented, "I think these vows are more useful than legal supervision. If you have no confidence in what you are selling, who would dare to swear like that? Hence I trust the food here".

*4.4. Food Heritagization and the Formation of a Sustainable Rural Tourism Destination*

4.4.1. Industrial Convergence and Economic Sustainability

The reproduction of traditional foods and its success has contributed to the emergence of other forms of tourism businesses since 2012, such as homestays and inns, bars, teahouses, creative studios, and parent–child activities. Notably, thanks to its reputation formed via tourism development, the village has managed to open snack restaurants in cities such as Xi'an and Xianyang since 2015, catering to those urbanite consumers in a more convenient way, further extending its food value-chain. Meanwhile, it also helps to promote the development of agricultural products processing businesses, such as flour mills and spicy workshops. This further boosts the demand for high-quality raw materials of agricultural products, which results in the emergence of vegetable plantations, animal breeding bases, and so on, to supply the materials. As a result, the village has realized the integrated development path of "the tertiary industry" driving "the secondary industry" and promoting "the primary industry" (Figure 6), seeing an industry convergence that greatly diversifies opportunities for local development [39]. In short, within less than ten years, food heritagization has fostered observable economic sustainability for the village.

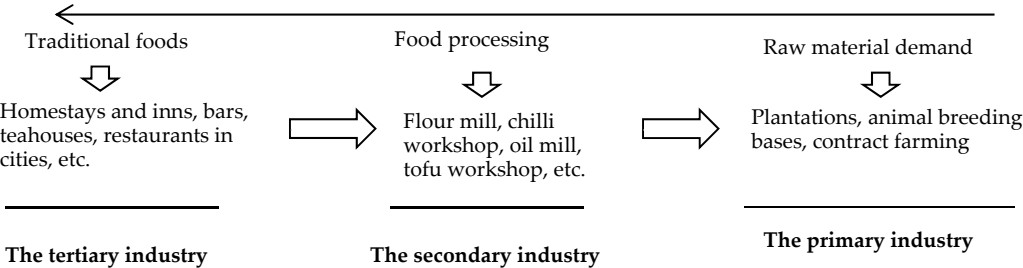

**Figure 6.** Industry convergence of Yuanjia Village.

4.4.2. Village Governance and Social Sustainability

Cooperatives: Enhancing Economic Equity

The success of traditional food reproduction at the village significantly boosted the income of food operators. "The sheep blood soup shop in the snack street earned at least RMB 3 million annually before 2014" (L-03). This, however, resulted in a huge income gap between food operators and ordinary villagers, and furthered psychological imbalance among ordinary villagers. Consequently, contradictions and conflicts began to emerge between ordinary villagers and food operators, and even among food operators themselves as business competition grew. As the Party secretary of the village (L-01) recalled, "When the business of the snack street got better, it also caused some problems, among which the growing income gap was the most serious one. Then I came up with the idea of setting up cooperatives to govern villagers and reduce the income gap".

In 2014, the village leaders decided to set up cooperatives in the Snack Street. However, for food operators, the establishment of cooperatives is tantamount to diluting their own income, hence they strongly backlashed the cooperative initiative at the beginning. "When the shop owners figured out what cooperatives were, they unanimously objected to letting others share their hard-earned money. Who would want that? And it is hard for peasants to see this matter in terms of long-term benefits" (L-02). But the village leaders insisted that only cooperatives can solve the problem of profit distribution, so cooperatives were enforced in the end.

With the gradual promotion of cooperatives, the tensions between ordinary villagers and food operators began to ease. For ordinary villagers, thanks to the cooperatives, although they do not need to participate in the daily operation, they can obtain a certain proportion of the profits from the snack street. For the operators, despite an income decrease after the introduction of cooperatives, they realized that their status in the village was significantly improving, and did not have to worry

about arousing the villagers' hatred. "At the end of each year, I am very happy when dividends are distributed. At the dividend distribution meeting, I can feel the enthusiasm of the villagers. Many villagers came to shake my hand and thanked me for sharing my profits. With the cooperatives, I earn less money than before, but I can sleep better at night, and I won't need to worry for the money I earn" (L-08). In fact, by early 2018, there were over 20 cooperatives at the village, such as a yogurt cooperative, a vermicelli cooperative, and a chili cooperative.

Peasant School: Enhancing Social Harmony in a Moral Way

Thanks to the rapid economic development of the village, many villagers got rich over a short period of time. Yet, some bad social phenomena came along with the fortunes, such as gambling, arrogance, and unhealthy comparisons with each other. As a response to these problems, the peasant school was built in the village to provide courses including "moral lectures", "recalling bitter memories", "Minglitang" (understanding the righteousness), etc. In this regard, one village official recollected, "In 2012, we found that some people would show arrogance after making more money, which had bad social impacts on the village. We then launched a 'recalling bitter memories' activity, in which everyone talked about the lives they lived before they came here [referring to operators from other villages], how much money they earned annually, and the lives they live now. This kind of reflection helps operators to think from a different perspective" (L-03).

In the process of tourism development, there have been many contradictions among villagers that cannot resort to law or governmental coordination. Instead, villagers or operators can resort to the peasant school to solve such problems via a moral way, which was popular back in the days. "When there is a conflict between villagers or commercial tenants, everyone goes to the moral lectures to discuss the situation. Once issues were discussed in the moral lectures, the problems are often naturally solved" (L-03). The endogenous channel has effectively promoted communication and unity among villagers and operators, contributing to social harmony.

## 5. Discussion

This study explores the mechanism of food heritagization and its impact on China's Yuanjia Village in the context of rural tourism. Led by local elites and monitored by a bottom-up regulatory system, locals use raw materials associated with being "local", "traditional", and "organic", make food with traditional crafts, and present food in a nostalgic atmosphere for consumption. Traditional foods are reinvented/reproduced as edible exemplars of the culture and heritage of the Guanzhong area, as well as carriers of nostalgia for a rural past that satisfies the imaginations and needs of surrounding urbanite visitors. This, in turn, contributes to the sustainability of the village as a rural tourism destination, featuring industry convergence that fosters economic sustainability, as well as governance embedded in rurality to deal with tourism benefit distribution (i.e., cooperatives) and social problems (i.e., peasant school) that promotes social sustainability.

While the findings unsurprisingly confirm the profound impact of Chinese urban consumption orientations and needs (i.e., nostalgia for rural life and a return to clean and healthy foods) on food heritagization on the ground [12], two things stand out from previous literature. First, locals play an active role in heritagizing local foods via manipulating the wider discourse on rurality and rusticity. Specifically, on one hand, the idyllically moral countryside, as the opposite of "corrupt" city [27], is highlighted to construct local foods produced by peasants as being safe, which may be best illustrated by its slogan "Farmers to Protect Food Safety". On the other hand, the backward countryside vis-à-vis the modern city, and its associated negative cultural connotations (e.g., dirty, absence of hygienic notions) [27] are largely avoided via embracing certain modern ways of food production, such as standardization of raw materials supply and hygiene measures. To this end, locals' food heritagization practices at Yuanjia village, to some extent, combats dominant "political discourses that define peasants and small-scale farming as China's agri-food 'problems' for which further capitalist industrialization is posed as the only and inevitable 'solution'" [40].

Second, self-organized governance embedded in rurality that has emerged during Yuanjia Village's food heritagization process effectively contributes to the sustainability of the village as a rural tourism destination. The dual structure of China's urban and rural areas means one must be aware of the fact that, unlike an urban China that is increasingly modernized and moving towards a Western way of life, rural China has its own distinctive nature, and rural Chinese lead their lives in a Chinese-specific manner [41]. Notably, rural China is very much so a guanxi-based society where traditions, kinship ties, community ties, etc. still play a salient role in people's social life [41,42], which may underpin the moral way of resolving social problems (i.e., peasant school). Likewise, peasant egalitarianism is also at play in rural China nowadays [43], which may account for the emergence of social contradictions, along with an income gap resulting from tourism development. In fact, similar phenomena are observed along with rural tourism development in previous studies, and a tourism benefit-sharing system to manage income gap among rural communities is argued as essential for long-term sustainability of rural tourism destination [44,45]. In particular, the success of food tourism cooperatives among locals echoes studies that advocate cooperatives as an alternative to capitalist industrialization for development in China's countryside [3,46].

## 6. Concluding Remarks

Unlike the previous studies focusing on macro-structural forces of Chinese state and market in shaping heritage foods [3,13,26], this study offers an on-the-ground empirical analysis of food heritagization. It contributes to the understanding of food heritagization from a bottom-up perspective as well as rural destination sustainability from a gastronomical perspective in China. Some practical implications can be obtained from this study. First, Chinese peasants are not passive actors to be modernized by state and/or capitalistic forces. Favorable policies should be made to encourage and facilitate local development initiatives and entrepreneurship [47]. Notably, the importance of rural elites in rural development should not be underestimated. Second, public discourse on food and rural areas should be closely monitored when food heritagization is adopted as a rural development strategy. This may help signal the distinctiveness of heritage foods to postmodern consumers. Third, previous studies have identified the high levels of fragmentation and lack of coordination of many farm and rural businesses as an obstacle for successful development of local food specialties [48,49]. This study suggests that bottom-up governance embedded in rurality (e.g., peasants' moral self-discipline, peasant development cooperatives) can help deal with such a problem to promote rural sustainability in China and similar contexts.

This research also has some implications for future studies. The current study is a place-specific research, albeit that its implications go beyond the case. One should remain cautious when generalizing the findings to other contexts. Further studies may examine the current results in different contexts, such as food tourism destinations in different geographical areas of China (e.g., ethnic minority areas in Western China), and destinations with different modes of food heritagization (e.g., top-down dominance by state forces and/or capitalistic enterprises). Moreover, different methods, including survey and experiments, might be adopted to investigate tourists' perceptions and experience of heritage foods from a multi-sensory perspective to generalize more practical guidance for heritage food production. Last but not least, food is closely related to identities [7,23,25]. In the current study, the foods are derived from local culture and tradition and refer to the distinctiveness of the area, be it the locally-produced materials, the traditional food making crafts, or the presentation of an atmosphere relating to a rural past of the area for food consumption. Accordingly, the foods are of relevance to locals' place identity. For urbanite visitors, the traditional foods are the carriers of a nostalgic rural "past" that has been losing along with the rapid urbanization and modernization in contemporary China, thus helping to confirm their current modern identity. Yet, the relationship between traditional foods and identities of related actors is not the focus of the current analysis. Further studies may tap into the significance of food heritagization for related actors on an individual level via an identity perspective.

**Author Contributions:** Conceptualization, J.G. (Jing Guan) and C.Z.; Formal analysis, J.G. (Jing Guan); Investigation, J.G. (Jing Guan); Methodology, J.G. (Jing Guan); Project administration, C.Z.; Supervision, C.Z.; Writing—original draft, J.G. (Jun Gao) and J.G. (Jing Guan); Writing—review and editing, J.G. (Jun Gao)

**Funding:** This research was supported by the National Natural Science Foundation of China (grant number 41471122).

**Conflicts of Interest:** The authors declare no conflict of interest.

**Note:** ① The vast majority of food operators are villagers from Yuanjia Village or surrounding villages. By "ordinary villagers", we mean these villagers who are not involved in food provision businesses. ② Zhanwu Guo was born in the village and is now the Party secretary of the village. He has largely led the development of Yuanjia village. ③ The food is made of vermicelli and sheep blood and is seasoned with pepper, chili oil, coriander, and so on. ④ The owner swears to promise that the food is made by hand in traditional ways and without any chemical additives; the flour, rapeseed oil, and seasonings are all provided by the workshop of Yuanjia Village. If one could not do that, he/she and his/her offspring would be punished by God.

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
