# Peer review of "Food Heritagization and Sustainable Rural Tourism Destination: The Case of China’s Yuanjia Village"

_sustainability, doi:10.3390/su11102858_

Round 1
Reviewer 1 Report
Dear author/s,
After I read your manuscript, which presents and interesting topic in the context of sustainable developmen, I have a few recommendations:
The introduction should present more specific the objectives of your research, not just the aim of the papaer.
the literature review should be extented also to other regions in order to emphasize the need and originality of your paper.
please mention how you select your respondents? Do you consider 36 respondents are representive for your research?
please mention the limitation of your study and future research directions. Which are the managerial implications?
Good luck!
Author Response
Thank you for your time and effort in reviewing our paper and offering constructive comments to us. Please find attached our point-by-point response to your comments.

Reviewer 2 Report
I very much liked reading the paper. I think the contribution might be of high importance towards identifying the social, environmental and economical sustainability of a rural tourism destination. Authors presented a case study village that has created a rural tourism destination based on local gastronomy. In fact i do not have major comments on an already well written paper.
My only major thought - question is the use of "heritagization" on the title and indeed as a concept used to support the paper's findings and field research. I am not so sure that the paper explores the socio-historical viewpoint- mechanisms of food heritagization in the village/area. We do not see any discussion on why this food used/provided by the food operators-in the tourism experience- is considered a heritage? How this food relates to local identity? What are the historical roots? Findings and discussion are not focused on roots of food as a marker of identity/heritage through a historical lens but rather present the whole spectrum of the development of rural (food) tourism in the village and the contribution of tourism activities to the whole economic activities (change in agriculture, etc).
To my reading of the paper, it is not about food heritage per se BUT rather on "Sustainable (economic, environmental, social) rural tourism development: the case of China's Yanjia Gastronomy village".
Some suggestions for the text:
L31: "It is not only about.." : what is? food provisioning?
L34: "..the study of (the relation between) food and tourism....."
L50-53: maybe rephrase -not clear enough (check grammar)
L86-87: ..."food gradually loses original taste, and may even ...."
Chapter 4.4.1 :it would be very interesting to add a timeline/comment -when did this change happen? after how many years?
[?] I am not so sure that i understood the origin of the food operators (that were involved in the conflict with the ordinary villagers)- are they not form the village themeselves? and in addition i am also not so sure about the use of the word "ordinary" for the locals. Maybe reconsider?
To make an overall comment..it would be very interesting to further discuss what were the success factors that "transformed" a rather "ordinary" as you put it village without tourism resources to the map of rural tourism destinations in terms of sustainability.
Author Response

(The authors gave the same response as above.)

Round 2
Reviewer 1 Report
Dear author/s,
Thank you for the improved version of your manuscript. I consider is suitable for being published now.
Good luck!
Author Response
Dear reviewer,
Thank you for you kind comments. We wish you all the best.
Kind regards,
Authors